# Theratyping of the Rare CFTR Genotype A559T in Rectal Organoids and Nasal Cells Reveals a Relevant Response to Elexacaftor (VX-445) and Tezacaftor (VX-661) Combination

**DOI:** 10.3390/ijms241210358

**Published:** 2023-06-19

**Authors:** Karina Kleinfelder, Valeria Rachela Villella, Anca Manuela Hristodor, Carlo Laudanna, Giuseppe Castaldo, Felice Amato, Paola Melotti, Claudio Sorio

**Affiliations:** 1Cystic Fibrosis Laboratory “D. Lissandrini”, Department of Medicine, Division of General Pathology, University of Verona, 37134 Verona, Italy; karina.kleinfelderfontanesi@univr.it (K.K.); carlo.laudanna@univr.it (C.L.); 2Department of Molecular Medicine and Medical Biotechnologies, University of Naples Federico II, 80138 Napoli, Italy; valeria.villella@gmail.com (V.R.V.); giuseppe.castaldo@unina.it (G.C.); felice.amato@unina.it (F.A.); 3CEINGE—Advanced Biotechnologies Franco Salvatore, 80145 Naples, Italy; 4Cystic Fibrosis Centre, Azienda Ospedaliera Universitaria Integrata Verona, 37126 Verona, Italy; ancamanuela.hristodor@gmail.com

**Keywords:** rectal organoids, CFTR modulators, nasal cells, rare mutations, cystic fibrosis, theratyping, Ussing chamber, personalized medicine, CFTR variants

## Abstract

Despite the promising results of new CFTR targeting drugs designed for the recovery of F508del- and class III variants activity, none of them have been approved for individuals with selected rare mutations, because uncharacterized CFTR variants lack information associated with the ability of these compounds in recovering their molecular defects. Here we used both rectal organoids (colonoids) and primary nasal brushed cells (hNEC) derived from a CF patient homozygous for A559T (c.1675G>A) variant to evaluate the responsiveness of this pathogenic variant to available CFTR targeted drugs that include VX-770, VX-809, VX-661 and VX-661 combined with VX-445. A559T is a rare mutation, found in African-Americans people with CF (PwCF) with only 85 patients registered in the CFTR2 database. At present, there is no treatment approved by FDA (U.S. Food and Drug Administration) for this genotype. Short-circuit current (Isc) measurements indicate that A559T-CFTR presents a minimal function. The acute addition of VX-770 following CFTR activation by forskolin had no significant increment of baseline level of anion transport in both colonoids and nasal cells. However, the combined treatment, VX-661-VX-445, significantly increases the chloride secretion in A559T-colonoids monolayers and hNEC, reaching approximately 10% of WT-CFTR function. These results were confirmed by forskolin-induced swelling assay and by western blotting in rectal organoids. Overall, our data show a relevant response to VX-661-VX-445 in rectal organoids and hNEC with CFTR genotype A559T/A559T. This could provide a strong rationale for treating patients carrying this variant with VX-661-VX-445-VX-770 combination.

## 1. Introduction

The monogenic, recessive disease cystic fibrosis (CF) is caused by mutations of the CF transmembrane conductance regulator (CFTR) gene that inhibits expression, activity, or trafficking of the CFTR anion channel at the apical surface of epithelial cells [1]. The impaired function of the CFTR channel leads to deficient salt and fluid transport that affects multiple organs, including the lung airways, pancreas, and sweat glands [2]. More than 85,000 individuals worldwide suffer from CF, the most common life-threatening disease in the Caucasian population [3]. CF presents variable clinical manifestations (including the pancreatic status: sufficient or insufficient) that are dependable on CF-causing variants, the environment and the genetic and epigenetic background of the patient [4]. Up to now, 401 CFTR mutations are known to be pathogenic while many of the over 2000 identified still lack information regarding their functional and clinical consequences (https://cftr2.org last accessed on 15 February 2023) and their response to available CFTR modulators. The classification of CFTR mutations into seven major groups, based on their effect on CFTR protein, simplified the prediction of clinical manifestation whilst supporting the selection of disease-modifying drugs [5,6]. Usually, patients with class IV–VI mutations have less severe CF phenotype whereas mutations belonging to class I, II and III typically produce the classical form of the CF disease [3,4,7]. Although this classification can be useful as guidance for defining the disease liability and treatment for the patient, CF patients can harbor modifiers genes or different CFTR genotypes, that cause combinatorial defects in the CFTR channel, which may contribute to the variable clinical response to available CF drugs [8,9].

The use of mutation-specific therapies now available for PwCF can be extended also for treating rare variants that have not been fully characterized, especially when tested in a close-to-native condition such as in primary cells derived from PwCF. The characterization of unclassified mutations is extremely important since it may help with the therapeutic decision-making for people carrying either non-F508del CF mutations or rare and uncharacterized pathogenic variants, including those that are excluded from clinical trials and are thus not candidates for receiving advanced therapy that may significantly improve their outcome.

In this study, we used both rectal organoids (colonoids) and primary nasal brushed cells (hNEC) derived from CF patient homozygous for A559T (c.1675G>A) variant to evaluate the responsiveness of this pathogenic variant to VX-770, VX-809, VX-661 and VX-661 combined with VX-445 (in a personalized medicine approach). We decided to use different tissues for assessing the restoration of CFTR function in order to provide the following: (1) additional evidence in support of the treatment and, (2) evaluation of CFTR response expressed in different anatomical districts [10]. A559T is a low-frequency mutation, found in North American black PwCF with only 85 patients registered in the CFTR2 database with this variant (https://cftr2.org last accessed on 15 February 2023). This missense mutation results in full-length CFTR that exhibits a strong decrease in the quantity of CFTR on the cell surface, due to defective NBD1-folding intermediates (categorized as class II) [11]. Previous reports in Fischer rat thyroid (FRT) cells suggest that A559T-CFTR is not responsive to VX-770, VX-661 and VX-661-VX-770 treatments [12]. Moreover, Zacarias and colleagues characterized the A559T-CFTR, expressed in the CFBE41o- cell line, as non-responsive to any market-approved CFTR modulators, through CFTR expression and energy prediction of ΔΔG of CFTR mutants based-algorithm analysis, lacking channel function studies [13]. At present, there is no treatment approved by FDA (U.S. Food and Drug Administration) for this mutation. Our results show that the A559T mutation affects CFTR protein maturation, since it fails to yield the fully glycosylated form of the protein (referred to as band C) and, thus, presents an absence of functional activity and data similar to results reported in other cell models [12,13,14]. Furthermore, our functional analysis demonstrates that A559T-CFTR activity can be substantially increased only in the presence of the combined use of correctors, VX-661-VX-445, as assessed in patient-derived rectal organoids and nasal epithelial cells. It is well known from the literature that even low levels of CFTR protein restoration can significantly rescue channel function [15,16]. Our CFTR expression analysis confirmed the partial recovery of fully glycosylated mutant CFTR protein upon double corrector therapy, suggesting a beneficial effect of VX-661-VX-445-VX-770 treatment for this CF patient.

## 2. Results

### 2.1. cAMP-Stimulated Anion Secretion for A559T-CFTR Variant Increases as Response to Combination Therapy, VX-661-VX-445, in Both Colonoids and Nasal Cells

We cultured nasal cells and intestinal organoids and performed transepithelial current measurement on A559T-CFTR rectal organoids grown as monolayers on transwells filters. CF F508del^+/+^ and non-CF (wild-type: WT/WT) colonoids were used as reference (Figure 1A,B). Following stimulation with forskolin (fsk 10 µM), A559T-colonoids elicited cAMP-dependent currents of 0.9 ± 0.2 µA/cm^2^ under control condition (DMSO; dimethyl sulfoxide), indicating a minimal function of CFTR, to a level that is similar to F508del/F508del colonoids (ΔIsc 1.2 ± 0.6 µA/cm^2^). These values represent less than 1% of wt-CFTR function as measured in our experimental settings. The augment registered for the short circuit currents for A559T-CFTR colonoids after pre-incubation with either VX-661 or VX-809 was insignificant: ΔIsc-VX-661, 1.5 ± 0.5 µA/cm^2^ and ΔIsc-VX-809, 1.0 ± 0.3 µA/cm^2^. The addition of potentiator VX-770 (0.3 µM) in acute and did not increase the Isc of chloride secretion for A559T-colonoids in line with previous data done in other A559T-CFTR expressing cell models [12] and at variance with the response recorded in corrected F508del-colonoids (Figure 1B). On the other hand, the triple therapy, VX-661-VX-445-VX-770, improved A559T-CFTR activity, increasing currents to 13.3 ± 4.4 µA/cm^2^ similar to Isc registered for VX-661-VX-445-VX-770 treated F508del/F508del colonoids (ΔIsc 13 ± 12 µA/cm^2^). The functional correction obtained for the A559T variant under the CFTR modulators combination correspond to approximatively 10% of WT-CFTR activity in our setting (Figure 1C,D). To assess the presence of synergistic effect between VX-661 and VX-445, we pretreated A559T colonoids with VX-445 alone for 20–24 h. VX-445 treatment increased the forskolin-activated short-circuit current (Isc) up to four-fold (ΔIsc 4 ± 2 µA/cm^2^) in comparison with the control group, representing 31% of VX-661-VX-445 rescue capacity registered in our settings (VX-661-VX-445-ΔIsc 12.8 ± 4.8 µA/cm^2^). Hence, VX-445 has an important synergistic effect on the restoration of A559T-CFTR processing and functional expression. These data demonstrate the potential effect of the combination of both correctors in recovering A559T-CFTR activity comparable to the improvements observed for F508del-CFTR colonoids treated with VX-661-VX-445-VX-770 (ΔIsc 20 ± 17 µA/cm^2^).

In order to reinforce the evidence and collect information on other tissues relevant to CF, we evaluated CFTR-dependent current in patient-derived nasal epithelia cells that were treated with the VX-661-VX-445-VX770 combination. A559T-CFTR function improvement reached a statistical significance following treatment with both correctors, with no significant effect for VX-770, coherent with results obtained in colonoids (Figure 2A,B).

### 2.2. Assessment of A559T-CFTR Protein Activity and Processing in 3D-Intestinal Organoids and under the Effect of CFTR Correctors

Forskolin-induced swelling (FIS) assay can also be used to indirectly measure CFTR activity and has been validated as a robust ex vivo biomarker and a good predictor of the clinical benefit of CFTR modulators [17]. We assessed CFTR activity after a 24 h incubation with VX-809, VX-661 and VX-661-VX-445. DMSO 0.1% (*v*/*v*) was the vehicle control. CFTR channel function was stimulated acutely with fsk and VX-770, as described in the figure legend (Figure 3A,B). The A559T-mutant rectal organoids presented a minimal function (AUC 126 ± 42) and showed no significant, drug-induced swelling for the monotherapy, VX-809 (AUC 146 ± 33), VX-661 (AUC 166 ± 47) and VX-770 (AUC 162 ± 38) after 120 min of stimulation with fsk (0.8 µM). Pre-treatment with VX-661 in combination with VX-445 elicited a strong CFTR activity, being noticed already at 0.02–0.128 µM fsk (AUC 431 ± 194), confirming the evidence derived from short-circuit current measurements. Of note is the observation that, at variance with the Ussing chamber assay, VX-770 appears to contribute to increase OG swelling when VX-661 is used in combination with VX-445, suggesting that under these culture condition VX-770 is likely more easily accessible to the cells. Worth mentioning, Spearman correlation test presented statistical significance between Isc data on colonoids and FIS done on 3D organoids at 0.8 µM forskolin, showing an excellent correlation between these two functional tests (Figure 3C).

The results of the functional assays cannot distinguish between channel hyper activation and expression level increase of a defective CFTR protein, since augment of CFTR channel function might be associated with an intensification of protein level, an enhancement of CFTR function or a combination of these two mechanisms. Investigating protein expression can help to better define the mechanism. We performed an immunoblot assay using anti-CFTR antibodies to assess the CFTR protein expression and maturation status/rescue of A559T-CFTR. F508del/F508del and non-CF (WT/WT) organoids were used here as a reference. A559T-CFTR presented a very low level of CFTR protein, expressed primarily as an immature form (core-glycosylated, band B) similar to what is seen for non-corrected F508del-CFTR protein. VX-809 or VX-661 treatment failed to improve the expression of the mature form of CFTR protein while the residual level of complex-glycosylated C band was detectable with VX-445 (Figure 4A,B). The band appearing over the C band does not represent a functional channel as Ussing chamber and FIS data indicate that its presence is not associated to a functional CFTR, and, thus, it is more probable that it is a nonspecific band. The double correctors combination, VX-661-VX-445, significantly increased the appearance of mature CFTR proteins, suggesting a capacity of the combined therapy to produce partial folding correction, in line with our functional analysis results. The signal associated to a fully glycosylated band of A559T-CFTR was similarly present in the F508del-CFTR variant following the same treatment. These data indicate that A559T-CFTR presents a folding defect that can be significantly rescued only by the VX-661-VX-445 combination (Figure 4A,B).

## 3. Discussion

Many CF-causing mutations remain incomplete and characterized functionally with new FDA approved drugs. Furthermore, the results of functional assays obtained by overexpression of CFTR variants in immortalized cells might not completely reflect the conditions found in primary cells, especially if derived exactly from the same patient under evaluation (precision medicine approach). It is well accepted that cell background interferes with the pharmacological rescue of mutant CFTR, since the data obtained from patient-derived cells is more reliable and thus of higher predictive value than those obtained from heterologous systems [18,19,20]. Recently, the FDA decided to expand the approval of VX-661-VX-445-VX-770 to additional CFTR variants that are responsive to this triple therapy in vitro, especially to mutations with a folding and/or trafficking defect (https://pi.vrtx.com/files/uspi_elexacaftor_tezacaftor_ivacaftor.pdf accessed on 12 December 2022). Here we have investigated the A559T-CFTR functional response to VX-770, VX-809, VX-661 and VX-445 alone or in combination, using two relevant ex vivo models: rectal organoids and nasal cells, providing reliable evidence of drug efficiency in restoring mutant CFTR function.

A559T is a pathogenic variant (also known as c.1675G>A) found in black PwCF [21,22,23] and it is associated with severe CF phenotype and pancreatic insufficiency (PI) [24,25]. This is a low-frequency mutation, having only three patients with A559T registered at CFTR-France (https://cftr.iurc.montp.inserm.fr/cftr, accessed on 10 March 2023) and 85 patients in total found at the CFTR2 database (www.cftr2.org, accessed on 10 March 2023). A559T is located in the coding exon 12 of the CFTR gene and results from G to A substitution (missense mutation) at nucleotide position 1675, replacing alanine at codon 559 to threonine [26]. These residues differ in polarity, size and charge and probably affect the secondary protein structure, disturbing the folding of the nucleotide binding domain one (NBD1) of CFTR [2,14,27].

Previous study demonstrated that the A559T-CFTR presents minimal function and it is not responsive in vitro to VX-809, VX-661, VX-770 and/or VX-661-VX-770 [28,29], in line with our findings. Moreover, in silico analysis demonstrates that A559T-related aberration cannot be overcome by using any of FDA-approved CFTR compounds [13]. Although molecular modeling and dynamics simulation [30] have been used to predict functional consequences of CFTR variants and the stability of mutant protein in presence of CFTR modulators, data from functional studies are still crucial for validating the data derived from in silico studies. In this context, tissue-derived samples allow us to study mutant CFTR function in response to CFTR modulators in patient-specific background and are considered the best available models to produce patient-clinically relevant information [17,31]. Our FIS and Ussing chamber records demonstrated a maximum increase of A559T-CFTR function upon VX-661-VX-445 combined treatment in both colonoids and nasal cells. Hence, to our knowledge, our work provides the first evidence of the effect of the modulator combination VX-661-VX-445 in improving the function of this variant in two different primary cell models derived from the same patient.

Western blot analysis revealed the presence of a maturation defect of A559T-CFTR protein, since the immature form (core-glycosylated, lower molecular weight) of CFTR was more easily detected in non-corrected intestinal organoids. This evidence confirms the previous findings that this variant belongs to the same class of F508del, i.e., class II, with severe folding defect [13,14,32,33]. Biochemical characterization done previously in cell lines have demonstrated that A559T-CFTR processing is not rescued by VX-809, VX-661 and VX-445 monotherapies, in agreement with our findings, and that VX-661-VX-445 combined treatment caused only modest, but insignificant, augmentation of the mature form of protein [13]. In our case, the double corrector therapy (VX-661-VX-445) was able to partially suppress the unfolding and premature degradation of A559T channel, suggesting that the combined action of VX-661 and VX-445 was important to provide a small restoration of A559T protein folding/trafficking sufficient to ensure a significant ion conductance. This is not surprising since other CFTR variants were already reported as better rescued with the use of combined therapy [19,34,35,36]. The little divergence encountered between our WB data and the previous work may indicate the presence of some difference in CFTR mutant’s response to CFTR modulators that may be dependent on the vitro model used. For instance, airway epithelial cells model may present altered gene expression, genetic instability, karyotype anomalies, due to immortalization and transformation process required to be suitable models for culturing and expansion [37,38,39], that may affect the CFTR variants recovery by the intervention of small molecules.

In our case, the addition of the type III corrector, VX-445, increased the abundance of the fully glycosylated protein (mature forms, higher molecular weight) and channel activity. Accordingly, the substantial recovery of CFTR-mediated anion secretion, registered in our functional tests, together with the partial rescue of A559T-CFTR folding defect is associated to a synergistic effect of the newly developed corrector VX-445 (elexacaftor) with VX-661 (tezacaftor). Indeed, VX-445 alone increased Isc currents, organoid swelling and protein expression level to a lower extent when compared to the combination where a synergistic effect is recorded. The combination facilitates the cellular processing and trafficking of mutant CFTR, which increases the amount of the CFTR protein delivered to the cell surface. This advantage is likely associated to the binding of these molecules to different sites on the CFTR protein. VX-661 (as well as VX-809), a type I corrector, stabilizes the interactions between NBD1 and the transmembrane domain one (TMD1) whereas, VX-445 further stabiles NBD1 and improves protein folding by interacting with TM helices 11, 2 and 10 and the lasso motif [34,36,40]. This strong pharmacological rescue capacity of VX-661-VX445 combination was previously demonstrated on primary bronchial epithelial cells derived from F508del homozygous patient [35,41], in our F508del-CFTR organoids and for other class II CF mutants [42,43].

## 4. Material and Methods

### 4.1. Biological Specimen Collection

Rectal biopsies were collected from CF participant with A559T/A559T (*n* = 1), F508del/F508del (*n* = 1) and WT-CFTR control participants (*n* = 2). Samples were taken during investigative or surveillance colonoscopy. Written informed consent was obtained from all participating subjects according to the local ethical committee’s rules (CRCFC-CFTR050)

### 4.2. Clinical Data

The CF patient of this study is a 32-years-old black male of central African origin, homozygous with the CFTR variant A559T. He had abnormal sweat Cl^−^ concentration at diagnosis: 119 ± 1 mmol/L; pancreatic insufficiency; chronic lung infection *Haemophilus* and *Achromobacter xilosoxidans* and impaired lung function (FEV1: 43%) of predicted value). Body Mass Index (BMI) (20.6) kg/m^2^.

### 4.3. Crypt Isolation and Organoid Culture from Human Rectal Biopsies

Crypts isolation and human intestinal organoids were obtained as described previously [44]. Briefly, human rectal biopsies recovered with colon forceps were immediately stored at 4 °C in surgical medium (RPMI-1640, glutamax 1X, HEPES 10 mM, P/S 1%, gentamicin 10 µg/mL and ciprofloxacin 20 µg/mL). Biopsies were then washed with cold PBS solution and incubated in 10 mM EDTA for 90–120 min at 4 °C. After washing, the isolated crypts were mixed with 50% Matrigel (Corning, Corning, NY, USA) and sown in 30 µL per well (with about 20–30 crypts/10 μL matrigel/droplet) in pre-warmed 24-well plates. After matrigel polymerization for 15–30 min at 37 °C, the plated crypts were covered with pre-warmed complete medium, consisting of 15% Advanced DMEM/F12 (supplemented with 1% penicillin and streptomycin, 0.2% primocin, 10 mM HEPES and 1% Glutamax), 1× N2, 1× B27 (all from Invitrogen), 1.25 mM N-acetylcysteine (Sigma, Kawasaki, Japan) and the following growth factors: 50 ng mL^−1^ mouse epidermal growth factor (mEGF; invitrogen), 50% Wnt3a-conditioned medium (WCM, Laguna Beach, CA, USA) and 10% noggin-conditioned medium (NCM), 20% Rspo1-conditioned medium, 10 mM nicotinamide (Sigma, Tokyo, Japan), 10 nM gastrin (Sigma), 500 nM A83-01 (Tocris, Bristol, UK), 1 μM SB 431542 (Tocris), 10 nM PGE (Sigma) and 3 μM SB202190 (Sigma). Additional antibiotics (Gentamycin and Vancomycin, Sigma, 1:1000 50 µg/mL) were used during the first week of culture. Medium was refreshed every other day and outgrowing crypts/organoids were expanded 1:3–1:5 times every 7–10 days. Complete medium was supplemented with 10 µM Rho inhibitor (Y27623) and 10 µM Chir (GSK3 inhibitor, CHIR-99021) (both from Sigma) during the first 2 days after seeding the crypts and after passaging.

### 4.4. Two-Dimensional Monolayer Culture

For the culture of epithelial monolayers, matrigel-embedded human rectal organoids were suspended in advanced DMEM/F12 (4 °C; Gibco) and washed by centrifugation (5 min, 200× *g*) to remove the Matrigel matrix. Intestinal organoids were dissociated by brief (75 s, 37 °C) incubation in trypsin (0.25%) solution (Gibco), followed by repeated (30×) aspiration through a 200 µL pipette tip. Dissociation was monitored by visual inspection (Olympus CKK31 inverted microscope), and the above procedure was repeated until most organoids had dissociated into small cell clusters. Trypsin activity was quenched by addition of fetal calf serum (10%) in advanced DMEM, and cells were washed in advanced DMEM and filtered through a cell strainer (70 µm; Falcon). Cells were counted using a hemocytometer and seeded (2.5 × 10^5^ cells/cm^2^) on permeable inserts (Transwell #3470; Corning) that had been pretreated with human placenta collagen, type IV (10 µg/cm^2^) (234154, Sigma-Aldrich, St. Louis, MI, USA) diluted in saline phosphate buffer and incubated at 37 °C for at least 2 h. Culture medium was the same used for extracellular matrix-embedded organoids, except that CHIR99021 (10 μM; Sigma-Aldrich, St. Louis, MI, USA) and Y-27632 (10 μM; Sigma-Aldrich) were added during the first two days after seeding. Cells were cultured until a confluent monolayer was obtained (7–14 days). For assessing the formation of a continuous epithelial monolayer, cultures were examined by microscopy and the transepithelial electrical resistance (TEER) was monitored using chopstick electrodes (EVOM2; World Precision Instruments, Sarasota, FL, USA).

### 4.5. TransEpithelial Electrical Resistance (TEER)

TEER was measured using an EVOM2 epithelial voltohmmeter (World Precision Instruments) before refreshing the medium. The readings of the voltohmmeter can be multiplied by the surface area of the Transwell inserts (0.33 cm^2^) to calculate the unit area of resistance (Ω·cm^2^). A TEER value of 400 Ω·cm^2^ was considered an index of complete monolayer formation.

### 4.6. Short-Circuit Current Recordings in Colonoids

Electrophysiological measurements were performed directly on the filter using specific Ussing chambers (P2300) and sliders (P2302T) (Physiologic Instruments, San Diego, CA, USA) and a voltage clamp EVC4000 (World Precision Instruments, Sarasota, FL, USA). For transepithelial current measurement, the colonocyte monolayers were incubated with ELX (VX-445, 3 μM; Med Chem Express, Monmouth Junction, NJ, USA), Luma (VX-809; 3 μM; Selleck Chemicals LLC, Houston, TX, USA) and TEZ (VX-661, 3 μM; Selleck Chemicals LLC, Houston, TX, USA), or vehicle (DMSO, 0.1%) for 20–24 h. Monolayers were bathed in symmetrical Meyler saline solution (pH 7.4) [10 mM Hepes; 0.3 mM Na_2_HPO_4_; 0.4 mM NaH_2_PO_4_; 1.0 mM MgCl_2_; 1.3 mM CaCl_2_; 4.7 mM KCl; 128 mM NaCl; 20.2 mM NaHCO_3_; 10 mM D-glucose] for the measurement of chlorine secretion mediated by CFTR. Solutions were maintained at 37 °C, gassed with 95% O_2_, 5% CO_2_. The transepithelial potential difference was clamped at 0 mV with a VVC-MC8 module (Physiologic Instruments), and the resulting short-circuit current (Isc) was recorded using a PowerLab 8/35 AD-converter (AD Instruments, Bella Vista, Australia) and associated software (LabChart v8; AD Instruments, Bella Vista, Australia). First, short circuit current reduction was blocked by 100 µM Amiloride (M) stimulus that inhibits the sodium channel EnaC; then filters were tested with components that act positively on CFTR activity: 10 µM forskolin (Sigma) applied to both apical (ap) and basolateral (bl) surfaces and 0.3 µM VX-770 (Selleckchem) (ap + bl). The experiment was concluded with the addition of the CFTR inhibitor, 20µM PPQ-102 (Tocris), from the apical and basolateral sides. At the end, 20 µM ATP (ap + bl) were just used to assess filters viability.

### 4.7. Nasal Brushing

Human nasal epithelial cells (HNEC) were sampled as reported [45]. Briefly, both nostrils were brushed and immersed in RPMI 1640 medium supplemented with 3% Penicillin-Streptomycin for cell culture. Then, each sample were incubated for 20 min in agitation at 600 rpm to collect all cells from brushes and their plating in a T12.5 collagen-coated flasks in PneumaCult–EX Medium (STEMCELLCat#: 05008), a serum free cell culture medium. Then, about 40,000 cells were seeded on porous filters (0.33 cm^2^, Transwell, Corning Cat#: 3470) in PneumaCult–EX Medium until confluence. The PneumaCultTM–EX Medium was replaced by PneumaCult-ALI Maintenance Medium (STEMCELL Cat#: 05022) for air-liquid interface (ALI) cultures.

### 4.8. Short-Circuit Current Recordings in hNEC

After at least 20 days of ALI culture, when the TEER value is up to 600 Ω·cm^2^, the function of epithelial nasal tissue was tested by Ussing Chambers system (Physiological Instruments). Both apical and basolateral hemi-chambers were filled with 5 mL of a solution containing (in mM): 126 NaCl, 0.38 KH_2_PO_4_, 2.13 K_2_HPO_4_, 1 MgSO_4_, 1 CaCl_2_, 24 NaHCO_3_, and 10 glucose, final pH 7–7.3. Both sides were continuously bubbled with a gas mixture containing 5% CO_2_, 95% O_2_ and the temperature of the solution was maintained at 37 °C. The transepithelial voltage was short-circuited with a voltage-clamp (VCC MC8 Multichannel Voltage/Current Clamp, Physiologic Instruments). The offset between voltage electrodes and the fluid resistance were canceled before experiments. The short-circuit current was acquired and analyzed using the Acquire and Analyze software Version 2.3.8.

### 4.9. Forskolin-Induced Swelling (FIS Assay)

Briefly, rectal organoids from a 7-day-old culture were seeded in a 96 well plate in 5 μL of 50% Matrigel (Corning) containing 30–40 organoids in 100 μL of culture medium with or without CFTR modulators: 3 µM VX-661 (Selleck Chemicals LLC, Houston, TX, USA), 3 µM VX-809 (Selleck Chemicals LLC, Houston, TX, USA) and 3 µM VX-445 (Med Chem Express) or combinations thereof. One day after seeding, organoid images were acquired at 37 °C and 5% CO_2_ humidified atmosphere in bright field at 5× magnification every 30 min, for a total acquisition of 120 min by using a wide field Zeiss AxioOberver 7 deconvolution microscopy setting (Carl Zeiss, Oberkochen, Germany). The microscope is equipped with Colibri 7 fluorescent LED illumination, motorized 3D scanning stage and ORCA-Flash4.0 V3 Digital CMOS camera (Hamamatsu Inc., Hamamatsu, Japan), set at an 8 output bit depth. We used an automatic Zen module for the time-lapse and processed with Zeiss ZEN 3.5. All conditions were analyzed in duplicate. The total organoid area (xy plane) increase relative to t = 0 of forskolin treatment was automatically quantified using the Carl Zeiss Zen 3.6 BioApp image analysis module, and normalized in respect to the initial value (100%). Normalized data were expressed as the total area under the curve (AUC, t = 120 min; baseline, 100%) calculated using GraphPad Prism version 7 (GraphPad Software, San Diego, CA, USA).

### 4.10. Immunoblotting

A559T/A559T, F508del/F508del and non-CF organoids were lysed in RIPA/EDTA/DTT/vanadate lysis buffer (50 mM Tris pH 7.5, 150 mM NaCl, 1% Triton X-100, 1% sodium deoxycholate, 0.1% SDS, 1 mM EDTA, 1 mM DTT, 1 mM sodium orthovanadate) with protease inhibitor cocktail (Roche, Mannheim, Germany) for 30 min in ice. Soluble fractions were analyzed by SDS-PAGE on 7.5% homemade Tris-Glycine gels. After electrophoresis, proteins were wet transferred from gel to a polyvinylidene difluoride (PVDF) membrane by electrophoresis with a transfer System (Bio-Rad, Hercules, CA, USA) at constant 100 V for 60 min. The membrane was blocked with 5% non-fat dry milk protein reconstituted in Tris-buffered saline–Tween (0.3% Tween, 10 mM Tris (pH 8) and 150 mM sodium chloride in water) and probed overnight at 4 °C with human CFTR-specific antibodies (cystic fibrosis folding consortium 450, 570 and 596, 1 in 1000 dilution in blocking solution). After the washing step, the membrane was probed with goat mouse-specific horseradish peroxidase (HRP)-conjugated secondary antibodies (Cell signaling, 1:12,000 dilution in blocking solution). As a loading control, we used β-Actin detected by α-beta-Actin antibody (1:1000) (Cell signaling, 4970). The blots were developed with ECL Westar Supernova ECL substrate (Cyanagen, Bologna, Italy). The imaging was performed using ImageQuantTM LAS 4000 software (GE Healthcare, Chicago, IL, USA) in a linear range of exposure. CFTR proteins level were quantified by densitometry of immunoblots using ImageJ Version 1.53t.

### 4.11. Statistical Analysis

Data are represented as mean ± S.D or mean ± SEM. GraphPad Prism 7.0 software (San Diego, CA, USA) was used for all statistical tests. Parametric or nonparametric test was used for comparison analysis between DMSO and treatment with CFTR modulators and *p*-values ≤ 0.05 were considered statistically significant.

## 5. Conclusions

In this study we demonstrated for the first time a significant effect of VX-661-VX-445-VX-770 treatment in correcting the A559T-CFTR folding defect and function on patient-derived rectal organoids and nasal cells, using two different functional assays: FIS (for intestinal OGs) and Ussing chamber (for both). Increased function is associated with an increased expression of fully processed CFTR protein (band C). The reported response to a specific combination of CFTR modulators may contribute to drug eligibility for all pwCF carrying at least one copy of this variant.

## Figures and Tables

**Figure 1 ijms-24-10358-f001:**
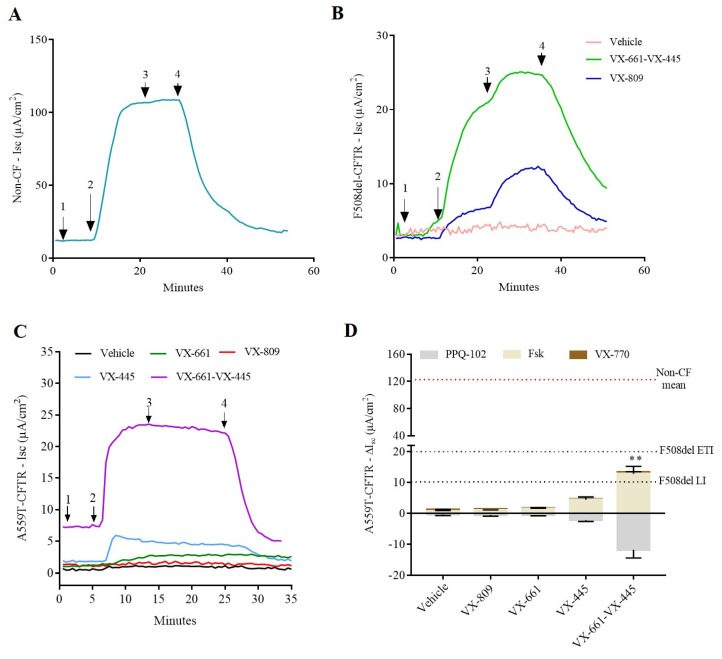
Electrophysiological response of 2D monolayer rectal organoids. (**A**) Representative transepithelial current measurements (TCM) in colonoids from non-CF (WT-CFTR) control. (**B**) Representative Isc tracing of chloride conductance mediated by CFTR in response to fsk and VX-770 for from a F508del/F508del subject whose rectal organoids were grown in Transwell^®^ filters. Cells were pre-incubated with DMSO (pink tracing) or VX-809 (blue tracing) and/or VX-661-VX-445 (green tracing). (**C**) Representative Ussing chamber recordings of transepithelial current measurements in A559T-colonoids pre-incubated with CFTR modulators (3 μM, 24 h, 37 °C) or vehicle (DMSO). Arrows indicate the addition of compounds: 100 µM apical amiloride (1. Amil), apical and basal addition of 10 µM forskolin (2. Fsk), apical and basal addition of 0.3 µM VX-770 (3. VX-770) and apical and basal addition of 20 µM CFTR-inhibition PPQ-102 (4. PPQ). (**D**) Isc response of cAMP-induced chloride currents by forskolin and augmented by VX-770 and CFTR modulators in rectal epithelial organoids of CF subject carrying the A559T variant. Values were normalized by the surface area of the filters in all the cases. Values are shown as mean (±SEM) of a minimum of three independent experiments, representing the maximum activation of A559T-CFTR colonoids after stimulation by fsk and VX-770 (0.3 µM). Kruskal-Wallis test ** *p* ˂ 0.005. The red dotted line indicates the mean value of currents registered for non-CF subjects (WT) in response to fsk. The blue dotted line indicates the mean value of currents registered for F508del-colonoids upon treatment with VX-445-VX-661-VX-770 (ETI) and the grey dotted line indicates the mean value of currents registered for F508del-colonoids upon treatment with VX-809-VX-770 (LI) both in response to fsk, used here as a reference.

**Figure 2 ijms-24-10358-f002:**
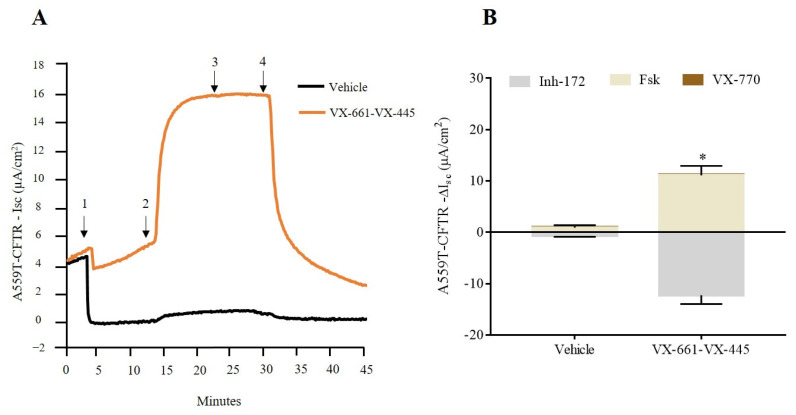
Functional improvement of triple corrector therapy on nasal epithelia derived from CF patient carrying the A559T/A559T variant. (**A**) Representative traces of vehicle (DMSO), and VX-661-VX-445 on nasal epithelial cells with the short-circuit current technique. Arrows indicate the addition of compounds: 100 µM apical amiloride (1. Amil), apical addition of 10 µM forskolin (2. Fsk), apical addition of 3 µM VX-770 (3. VX-770) and apical addition of 10 µM CFTR-inhibition Inh-172 (4. Inh-172). (**B**) Isc response of cAMP-induced chloride currents by fsk and augmented by VX-770 and CFTR modulators in nasal cells of CF subject carrying A559T mutation. Values were normalized by the surface area of the 2D filters and are shown as mean (±SEM) of three independent experiments. Mann-Whitney test * *p* ≤ 0.05.

**Figure 3 ijms-24-10358-f003:**
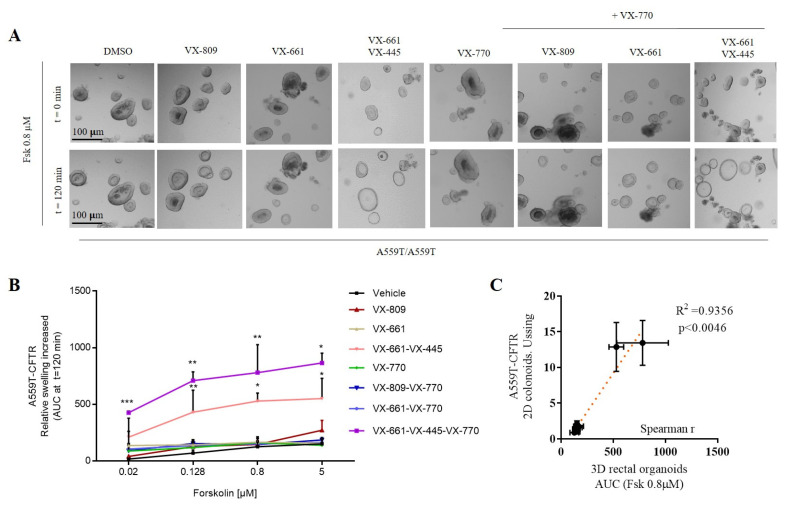
FIS rates in A559T/A559T-organoids. (**A**) Bright-field microscopy images treated organoids in response to fsk (0.8 µM) alone or together with VX-770 (3 µM). (**B**) Swelling of rectal organoids induced by fsk alone or in combination with VX-770 from two independent experiments. Data are expressed as the absolute Area Under the Curve (AUC) of each duplicate for a fsk dose from 0.02 µM to 5 µM and calculated from time tracings comparable to baseline (100%, t = 120 min). Representative statistical significance (*t* test). Data are means ± SD of two independent experiments. Asterisks indicate significant difference compared with vehicle (* indicates *p*-value < 0.05, ** *p*-value < 0.005, *** *p*-value < 0.0005 one-way ANOVA). (**C**) Spearman correlation of fsk-stimulated current (Isc) versus FIS assay in DMSO, VX-770, VX-809, VX-661 and VX-661-VX-445 treated rectal organoids. The average values of response to therapies measured from rectal organoids swelling at 0.8 µM forskolin were matched with average values of short-circuit measurements (DMSO-, VX-770-, VX-809-, VX-661- and VX-661-VX-445-treated organoids). R-Spearman’s rank correlation coefficient and associated *p*-value, showing an excellent correlation between the two functional assays.

**Figure 4 ijms-24-10358-f004:**
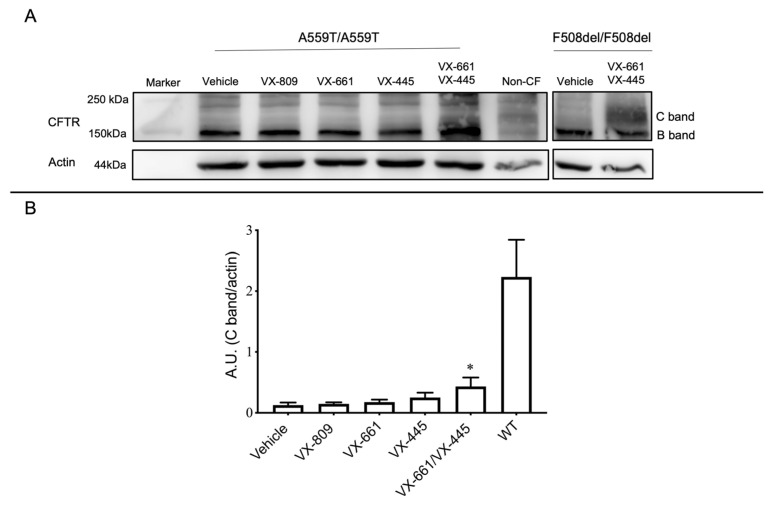
Western blot analysis of CFTR and β-actin (loading control) expression in rectal organoids. (**A**) Representative A559T-CFTR in colonoids after 24-h pre-treatment with vehicle (DMSO 0.1%), VX-809 (3 µM), VX-661 (3 µM), VX-445 (3 µM) alone or together with VX-661 (3 µM). C band: mature, complex-glycosylated CFTR; B band: immature, core-glycosylated CFTR. F508del-CFTR protein expression corrected with VX-661-VX-445 and non-CF (WT-CFTR) were used as reference. (**B**) Quantification showing a substantial increase in the A559T protein expression of mature form of CFTR after treatment with a combination of VX-661 and VX-445 correctors in rectal organoids. WT-CFTR control was used here as reference, vales were normalized with actin levels. Calculation performed used β-actin as the loading control. Data are mean ± SEM of a minimum of three independent experiments, * indicates *p*-value < 0.05 one-way ANOVA.

## Data Availability

Not applicable.

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
