# Peer review of "Theratyping of the Rare CFTR Genotype A559T in Rectal Organoids and Nasal Cells Reveals a Relevant Response to Elexacaftor (VX-445) and Tezacaftor (VX-661) Combination"

_ijms, 2023, doi:10.3390/ijms241210358_

Round 1

Reviewer 1 Report

Kleinfelder, et al. assessed the impact of elexacaftor-tezacaftor combination on the functional activity and processing of A559T CFTR in rectal organoids and nasal cells from a patient homozygous for this mutation. The authors found that the combination of these two correctors provided statistically significant improvement in CFTR function, resulting in over 10% of activity of normal individuals. A moderate increase in CFTR processing was also observed as a result of treatment with the dual corrector combination. This is an excellent example of utilization of combination of known correctors to treat CF patients with rare mutations. The research was well conducted in most parts. I only have a few minor comments.

1) The processing of CFTR could be better demonstrated by better Western blots. In most cases, the band C is invisible, making the quantification less reliable. If possible, could the authors include a blot for F508del CFTR treated with VX-809 as yet another control?

2) The term "rector" is a bit odd. It would be better to use the standard terms such as correctors and potentiators. Specifically, on page 3, line 99, the authors should have used the term "triple modulator combination" instead of "triple corrector combination" as VX-770 is not a corrector.

3) Page 3, line 104, the demonstration of synergy should include the impact of both elexacaftor and tezacaftor when used alone.

Minor language editing is needed in several places.

Author Response

Kleinfelder, et al. assessed the impact of elexacaftor-tezacaftor combination on the functional activity and processing of A559T CFTR in rectal organoids and nasal cells from a patient homozygous for this mutation. The authors found that the combination of these two correctors provided statistically significant improvement in CFTR function, resulting in over 10% of activity of normal individuals. A moderate increase in CFTR processing was also observed as a result of treatment with the dual corrector combination. This is an excellent example of utilization of combination of known correctors to treat CF patients with rare mutations. The research was well conducted in most parts. I only have a few minor comments.

1) The processing of CFTR could be better demonstrated by better Western blots. In most cases, the band C is invisible, making the quantification less reliable. If possible, could the authors include a blot for F508del CFTR treated with VX-809 as yet another control?

We performed another WB and replaced the previous WB images, including VX-809 as shown in the new fig 4.

2) The term "rector" is a bit odd. It would be better to use the standard terms such as correctors and potentiators. Specifically, on page 3, line 99, the authors should have used the term "triple modulator combination" instead of "triple corrector combination" as VX-770 is not a corrector.

 Agree, we have modified the terminology (line 112).

3) Page 3, line 104, the demonstration of synergy should include the impact of both elexacaftor and tezacaftor when used alone.

Done.

Reviewer 2 Report

Comments to Authors

General comments -

1.       The patient description providing the biospecimens is detailed, see section 5.2, and may be identifying.

2.       Figure 1 provides investigations of control samples only involving non-CF and F508del homozygous genotypes, whereas Figure 2 gives actual results of sample of the genotype under investigation. Currently, Figures 1 and 2 have identical headings/titles and should be combined for efficiency of space and content. (Panels Figure 1A, Figure 1B and Figure 2A present measurements using common tools and analyses.)

3.       What is the interpretation of the absence of effect (Figure 3, panel A for VX-445 + VX-661 treatment) and current drop evident (Figure 2A for VX-445 and VX-445 + VX-661) upon inclusion of ivacaftor (VX-770) in the rectal organoid and nasal epithelia culture assays, respectively?  

4.       It is unclear how the correlation indicated in Figure 4C was determined, as there are only two discernable measures in the figure.  Could the additional contributing data sets be shown expanded in an inset to clarify and help understand how the correlation was obtained. 

5.       The functional investigations support a modest restoration of CFTR activity with drug combinations of VX-661, VX-445 and VX-770 in cultured rectal organoid and nasal epithelial cells with homozygosity for the c.1675G>A CF-causal variant.  There is also support from protein analyses, but this supporting data should be more clearly explained in both the Results and Discussion sections. 

Specifically, regarding Figure 5:

i)     How much Band C was achieved compared to rescued F508del/F508del, how does this align with the functional assessments of the respective genotypes?

ii)    Band C labeling for the A559T/A559T sample is not clearly aligned across the samples indicated in either Panels A or B.  Were electrophoresis conditions consistent? It is not clear that inclusion of modulator treatment leads to increased levels of Band C, as Band B also appears to vary with modulator treatments?

iii)  What is the larger migrating band relative to the indicated Band C (glycosylated CFTR) in the A559T/A559T samples, is this a non-specific band?   

Typographical and other comments -

1.       The presentation of the Figures could be improved, with more consistent matching of panels with respect to description and font sizing. (The font sizing of most panels should be increased and be consistent within a Figure.

2.       The text and Figure 2 legend refers to A and B panels, but the panels are not labelled in the Figure.

3.       The indicated dotted blue and grey lines in current Figure 2B are not consistent with the legend description. 

4.       What is ‘www.cfww.org’ on line 43, page 2?

5.       ‘Not corrected’ on line 243, page 7 should read ‘non-corrected’.

6.       ‘10um Cihr, line 307, page 9 should read ‘10uM GSK3 inhibitor’?

7.       ‘human placenta’ on line 320, page 9 should read ‘human placenta Collagen, type IV’

8.       ‘in both sides … and’ on line 351, page 9 should read ‘applied to both apical (ap) and basolateral (bl) surfaces and’.

Acceptable, with correction of minor typographical and spelling errors.

Author Response

  1. The patient description providing the biospecimens is detailed, see section 5.2, and may be identifying.

This is a common issue for exceedingly rare variants and demographic information are important to describe the case with details useful to the clinicians. However as the precise country of origin does not represent a key information we propose to mention “central Africa” origin instead of Ghana.

  1. Figure 1 provides investigations of control samples only involving non-CF and F508del homozygous genotypes, whereas Figure 2 gives actual results of sample of the genotype under investigation. Currently, Figures 1 and 2 have identical headings/titles and should be combined for efficiency of space and content. (Panels Figure 1A, Figure 1B and Figure 2A present measurements using common tools and analyses.)

         We agree and combined figures 1 and 2

  1. What is the interpretation of the absence of effect (Figure 3, panel A for VX-445 + VX-661 treatment) and current drop evident (Figure 2A for VX-445 and VX-445 + VX-661) upon inclusion of ivacaftor (VX-770) in the rectal organoid and nasal epithelia culture assays, respectively?

        The drop of current after the acute addition of potentiator, VX-770, suggest that the presence of this molecule disturbed the signal registered. The interpretation of the absence of the effect after adding VX-770 is that this molecule does not further improve A559T-CFTR function in Ussing technique when used alone or that we observed for VX-661-VX-445 treated colonoids on nasal cells a ceiling effect that does not allow us to appreciate the effect of VX-770. For better interpretation, we replace the previous tracing for rectal organoids to one that presents less disturbance after VX-770 addition. Please allow a technical note: VX770 is notoriously highly hydrophobic (Csana ́ dy and To ̈ro ̈csik. eLife 2019;8:e46450. DOI: https://doi.org/10.7554/eLife.46450) and its solubility in Ussing chamber conditions makes it a difficult reagent to handle, while in FIS assay the response is usually more reliable.

  1. It is unclear how the correlation indicated in Figure 4C was determined, as there are only two discernable measures in the figure. Could the additional contributing data sets be shown expanded in an inset to clarify and help understand how the correlation was obtained.

        The correlation compares the mean values obtained from FIS assay and Ussing technique for each condition tested, i.e., mean values for control group, VX-809, VX-661 and VX-661-VX-445. Hence, it checks how linear was the distribution of the values obtained after treating A559T organoids with different CFTR modulators using two different functional analysis. Our result demonstrates that the both functional assays used by us showed the same message: A559T-CFTR activity recovery under VX-661-VX-445 treatment. We included additional information regarding this graphic on the legend.

  1. The functional investigations support a modest restoration of CFTR activity with drug combinations of VX-661, VX-445 and VX-770 in cultured rectal organoid and nasal epithelial cells with homozygosity for the c.1675G>A CF-causal variant. There is also support from protein analyses, but this supporting data should be more clearly explained in both the Results and Discussion sections. 

Specifically, regarding Figure 5:

  1. i)     How much Band C was achieved compared to rescued F508del/F508del, how does this align with the functional assessments of the respective genotypes?

We have quantified the recovery in comparison with WT CFTR, now shown in new figure 4.

  1. ii)    Band C labeling for the A559T/A559T sample is not clearly aligned across the samples indicated in either Panels A or B. Were electrophoresis conditions consistent? It is not clear that inclusion of modulator treatment leads to increased levels of Band C, as Band B also appears to vary with modulator treatments?

We replaced the WB images with a new one in which all conditions tested for A559T-CFTR is present in the same blot and protein marker is also clearly detectable. For WT signal, we decided to load less amount of protein to facilitate the recognition of C- and B bands when comparing with A559T-CFTR.

iii)  What is the larger migrating band relative to the indicated Band C (glycosylated CFTR) in the A559T/A559T samples, is this a non-specific band?  

This band appear in all the condition tested but does not constantly. As it is present also in condition where no FSK elicited currents are detected, this fact (together with the higher MW with respect to C band detected in the control samples) suggest that this represent a non-specific band.

Typographical and other comments :

  1. The presentation of the Figures could be improved, with more consistent matching of panels with respect to description and font sizing. (The font sizing of most panels should be increased and be consistent within a Figure.

          Thanks. Done

  1. The text and Figure 2 legend refers to A and B panels, but the panels are not labelled in the Figure.

Thanks, we modified the figure 2 and as suggested.

  1. The indicated dotted blue and grey lines in current Figure 2B are not consistent with the legend description.

Thanks, we corrected the information regarding dotted blue and grey lines.

  1. What is ‘www.cfww.org’ on line 43, page 2?

We decided to remove this site on line 43 because it has been permanently disolved. Thanks

  1. ‘Not corrected’ on line 243, page 7 should read ‘non-corrected’.

           Done.

  1. ‘10um Cihr, line 307, page 9 should read ‘10uM GSK3 inhibitor’?

          Corrected to 10 µM Chir (GSK3 inhibitor). Thanks.

  1. ‘human placenta’ on line 320, page 9 should read ‘human placenta Collagen, type IV’

         Done. Thanks

  1. ‘in both sides … and’ on line 351, page 9 should read ‘applied to both apical (ap) and basolateral (bl) surfaces and’.

         Done. Thanks

Comments on the Quality of English Language

Acceptable, with correction of minor typographical and spelling errors.

Reviewer 3 Report

1.     The introduction needs improvement and should conclude with a summary of the study's results and significance.

2.     On page 2, lines 75-77, it appears that reference 12 may not be the correct reference for previous reports.

3.     Previous studies have shown that A559T-CFTR does not improve CFTR expression or function with any CFTR modulators (doi.org/10.3390/ijms24043211). However, the present study found it to be responsive. The reasons for the discrepancy in results should be discussed.

4.     The A559T-CFTR evaluation was performed on cells/organoids derived from only one patient, and it is possible that individual differences may exist. Therefore, evaluating at least one more patient-derived cell/organoid is necessary.

5.     In Fig. 2 and Fig. 3's right panel, the sample size is unknown for the quantitative data, and it needs to be shown.

6.     In Fig. 4A, a scale bar needs to be added to the images.

7.     In Fig. 5, protein markers (size markers) need to be added to the immunoblotting data.

8.     The immunoblotting data in Fig. 5 is not convincing. The authors should consider using endo H and/or PNGase F digestion to clarify whether the CFTR bands are mature forms or not.

Author Response

  1. The introduction needs improvement and should conclude with a summary of the study's results and significance.

Done. We included the summary of the study and its significance at the end of the introduction.

  1. On page 2, lines 75-77, it appears that reference 12 may not be the correct reference for previous reports.

Agree, we have modified the text and reference

  1. Previous studies have shown that A559T-CFTR does not improve CFTR expression or function with any CFTR modulators (doi.org/10.3390/ijms24043211). However, the present study found it to be responsive. The reasons for the discrepancy in results should be discussed.

Done. Thanks. We included the mentioned study in our work and we discussed it, indeed the data regarding protein expression are obtained in different model systems and cell type-dependent processing is expected to differ. Our data are obtained in two primary cells models from the same subject while the study mentioned by the reviewer is done in cDNA transfected CFBE cells and mostly overlap at the protein expression level where we only find a significant difference following VX661-445 treatment.

  1. The A559T-CFTR evaluation was performed on cells/organoids derived from only one patient, and it is possible that individual differences may exist. Therefore, evaluating at least one more patient-derived cell/organoid is necessary.

We would be delighted to do so however the rarity of the genotype impose to communicate the results of only this case, as no other subjects beside this one are available for us. We are following the N-of-1 case study approach that apply to these exceedingly rare cases, and we underline the fact that the results are obtained in two independent cell models from the same subject increasing the significance of the finding. Other researchers and clinicians  might find these results useful and might have the opportunity to report further cases. 

  1. In Fig. 2 and Fig. 3's right panel, the sample size is unknown for the quantitative data, and it needs to be shown.

We wrote the number of replicates done in the legend.

  1. In Fig. 4A, a scale bar needs to be added to the images.

Done.

  1. In Fig. 5, protein markers (size markers) need to be added to the immunoblotting data.

Done.

  1. The immunoblotting data in Fig. 5 is not convincing. The authors should consider using endo H and/or PNGase F digestion to clarify whether the CFTR bands are mature forms or not.

We believe that the emergence of  new bands corresponding to a partially processed antigen is clearly and reproducibly detectable, although we agree that it is not as evident as for F508del corrected case where a more uniform C band is visible. However this has to be considered a feature of this variant. We want to remark the fact that we did not rely only on this evidence to describe the response to modulators even if, as reported by other authors, even a very low evidence of CFTR protein expression can be associated to a strong functional response (see ref. 15 and 16). We believe that the combination of WB and functional data in two cell models are strong enough to support our conclusion, that has a clear relevance for the clinicians in charge of the case described (and possibly other similar), especially in the presence of another published report that describe absence of response to the same combined treatment. However this study is  based only of protein expression data obtained on transfected cell lines and is in line with our findings in all the other conditions tested. As for the enzymatic treatment unfortunately PNGaseF  is still in backorder from the company and, given the extended deadline, we decided to resubmit the work to your attention as we believe that you might agree that this experiment, although of some interest,  is not critical for sum of  reasons mentioned above.

Round 2

Reviewer 3 Report

Thank you for making the adjustment to the manuscript. Now I am happy to accept your manuscript for publication in IJMS after a minor correction. In Figure 4, there is inconsistency in the molecular weight size notations (250 kDa, 150kDA, 44kDA). Please correct it.

Author Response

Dear Reviewer, indeed there was a typing error we missed, thank you for having spot it.